# Human Tumor–Derived Matrix Improves the Predictability of Head and Neck Cancer Drug Testing

**DOI:** 10.3390/cancers12010092

**Published:** 2019-12-30

**Authors:** Katja Tuomainen, Ahmed Al-Samadi, Swapnil Potdar, Laura Turunen, Minna Turunen, Piia-Riitta Karhemo, Paula Bergman, Maija Risteli, Pirjo Åström, Riia Tiikkaja, Reidar Grenman, Krister Wennerberg, Outi Monni, Tuula Salo

**Affiliations:** 1Department of Oral and Maxillofacial Diseases, Clinicum, Faculty of Medicine, University of Helsinki, 00014 Helsinki, Finland; katja.tuomainen@helsinki.fi (K.T.); ahmed.al-samadi@helsinki.fi (A.A.-S.); minna.turunen@helsinki.fi (M.T.); 2Translational Immunology Research Program (TRIMM), University of Helsinki, 00014 Helsinki, Finland; 3Institute for Molecular Medicine Finland (FIMM), University of Helsinki, 00290 Helsinki, Finland; swapnil.potdar@helsinki.fi (S.P.); laura.turunen@helsinki.fi (L.T.); krister.wennerberg@helsinki.fi (K.W.); 4Research Programs Unit, Genome-Scale Biology Program and Medicum, Biochemistry and Developmental Biology, University of Helsinki, 00014 Helsinki, Finland; piia-riitta.karhemo@helsinki.fi (P.-R.K.); outi.monni@helsinki.fi (O.M.); 5Biostatistics Consulting, Department of Public Health, University of Helsinki and Helsinki University Hospital, 00014 Helsinki, Finland; paula.h.bergman@helsinki.fi; 6Cancer and Translational Medicine Research Unit, University of Oulu, 90014 Oulu, Finland; maija.risteli@oulu.fi (M.R.); pirjo.astrom@oulu.fi (P.Å.); riia.tiikkaja@student.oulu.fi (R.T.); 7Department of Otorhinolaryngology—Head and Neck Surgery, Turku University Hospital, University of Turku, 20520 Turku, Finland; reigre@utu.fi; 8Biotech Research and Innovation Center, Department of Health and Medical Sciences, University of Copenhagen, 2200 Copenhagen, Denmark; 9Medical Research Center, Oulu University Hospital, 90014 Oulu, Finland; 10Helsinki University Hospital, 00029 Helsinki, Finland

**Keywords:** head and neck cancer, drug screening, human tumor microenvironment, clinical trials, in vitro 3D

## Abstract

In vitro cancer drug testing carries a low predictive value. We developed the human leiomyoma–derived matrix “Myogel” to better mimic the human tumor microenvironment (TME). We hypothesized that Myogel could provide an appropriate microenvironment for cancer cells, thereby allowing more in vivo–relevant drug testing. We screened 19 anticancer compounds, targeting the epidermal growth factor receptor (EGFR), MEK, and PI3K/mTOR on 12 head and neck squamous cell carcinoma (HNSCC) cell lines cultured on plastic, mouse sarcoma–derived Matrigel (MSDM), and Myogel. We applied a high-throughput drug screening assay under five different culturing conditions: cells in two-dimensional (2D) plastic wells and on top or embedded in Matrigel or Myogel. We then compared the efficacy of the anticancer compounds to the response rates of 19 HNSCC monotherapy clinical trials. Cancer cells on top of Myogel responded less to EGFR and MEK inhibitors compared to cells cultured on plastic or Matrigel. However, we found a similar response to the PI3K/mTOR inhibitors under all culturing conditions. Cells grown on Myogel more closely resembled the response rates reported in EGFR-inhibitor monotherapy clinical trials. Our findings suggest that a human tumor matrix improves the predictability of in vitro anticancer drug testing compared to current 2D and MSDM methods.

## 1. Introduction

Current preclinical anticancer drug testing carries a low predictive value, since only 5% of compounds showing efficacy in in vitro tests are licensed following clinical trials [1]. Therefore, improved predictive in vitro methods are urgently needed. We developed a human tumor leiomyoma–derived matrix “Myogel” to better mimic the human tumor microenvironment (TME) [2,3]. Myogel provides a three-dimensional (3D) culture environment for in vitro cancer studies, preserving the soluble factors present in the human TME by including, for example, soluble cytokines and growth factors [3]. Myogel proteome differs greatly from mouse sarcoma–derived Matrigel [3]. We previously demonstrated that 66% of the Myogel protein content differs from Matrigel [3]. Yet, these matrices share several proteins such as laminin, type IV collagen, heparan sulfate proteoglycans, nidogen, and epidermal growth factor [3]. Carcinoma cells show a markedly higher migration and invasion ability on Myogel compared to Matrigel [2,3,4].

Head and neck squamous cell carcinoma (HNSCC) represents the sixth most common cancer worldwide, with a relatively low survival rate of around 50% [5]. Treatment approaches include surgery combined with radio-, chemo-, or epidermal growth factor receptor- (EGFR-) targeted therapy relying on cetuximab (Erbitux). EGFR is frequently overexpressed in HNSCC [6] and its inhibitors have shown clear anticancer effects in both in vitro and in vivo preclinical studies [7,8,9,10,11]. However, most EGFR inhibitors have yielded disappointing results in HNSCC clinical trials and there are no clinically approved predictive biomarkers for EGFR inhibitor response [12]. We propose that the presence of the human tumor matrix, as a physiologically relevant extracellular matrix for human cancer cells, could improve the predictability of anticancer drug testing in preclinical studies of HNSCC.

In this study, using a high-throughput drug screening method, we tested 12 HNSCC cell lines cultured on plastic, mouse sarcoma–derived Matrigel, or Myogel using 19 anticancer compounds targeting EGFR, MEK, and PI3K/mTOR. We compared the efficacy of the anticancer drugs tested in vitro against the patients´ response rates in 19 monotherapy clinical trials. We identified different drug effects between Myogel and the other testing conditions, and demonstrated that Myogel in in vitro drug screening replicates the best results from clinical trials. 

## 2. Results

### 2.1. In Vitro Drug Screen

The cell viability readout revealed differential EGFR- and MEK-inhibitor effects between Myogel and the other testing conditions (Figure 1A,B and Appendix A). However, cells responded similarly to PI3K/mTOR inhibitors independent of the culturing condition (Figure 1C and Appendix A). Based on previous studies, drug response profiles were divided into four activity classes using the drug-sensitivity score (DSS) values and artificial cut-off points: inactive DSS < 5, low 5 ≥ DSS < 10, moderate 10 ≥ DSS < 15, and high DSS ≥ 15 [13]. Based on the DSS values, the EGFR and MEK inhibitors showed a moderate or high activity in most cell lines cultured on plastic, and on top of or embedded in Matrigel (Figure 1A,B and Appendix A). However, most cell lines cultured on top of or embedded in Myogel showed a low EGFR- and MEK-inhibitor activity or no effect (Figure 1A,B and Appendix A). Collectively, cells responded relatively similarly to the PI3K/mTOR and mTOR inhibitors across the different culturing conditions (Figure 1C and Appendix A). mTOR inhibitors (Rapalogs: temsirolimus and ridaforolimus) showed a relatively low activity level on most cells lines (Appendix A). In addition, everolimus was inactive in most cell lines (Appendix A). The PI3K/mTOR inhibitor omipalisib showed a moderate-to-high activity level across all cell lines and under the different conditions (Appendix A). Only one cell line (UT-SCC-24B) responded to dactolisib (Appendix A). Sirolimus and PF-04691502 showed varying a low-to-high activity level across cell lines (Appendix A). Furthermore, apitolisib exhibited a low-to-moderate activity (Appendix A). 

We performed the Friedman test followed by a pair-wise comparison of drug responses under five culturing conditions (control, Myogel two-dimensional (2D), Myogel 3D, Matrigel 2D, and Matrigel 3D) for all cell lines in order to detect differences in drug efficacy [14]. Significantly different responses between culturing conditions were observed for several drugs, particularly for the EGFR and MEK inhibitors (Table 1). Significant differences were consistently seen between Myogel 3D and Matrigel 3D in the EGRF and MEK inhibitors (Table 1). The MEK inhibitor responses were significantly diminished in Myogel compared to Matrigel (100% of all cases). The PI3K/mTOR inhibitors, omipalisib, apitolisib and PF-04691502, were significantly less effective in Myogel 2D and 3D compared with Matrigel 3D (Table 1). Although the mTOR inhibitor everolimus exhibited a generally weak response, it showed a higher effect in Myogel 2D and 3D compared with Matrigel 3D (Table 1 and Appendix A). The remaining PI3K/mTOR along with the selective mTOR inhibitors (temsirolimus, sirolimus, ridaforolimus, and dactolisib) showed no significant differences.

### 2.2. Clinical Trial Data Collection

We collected data from clinical trials on the anticancer compounds used as monotherapies for HNSCC patients until 17 March 2018 (Table 2). Only cetuximab, erlotinib, afatinib, and gefitinib were tested in phase III trials. Erlotinib has been widely studied, but only as a combination therapy with other treatments. No clinical trials have been performed among HNSCC patients for the following drugs: canertinib (EGFR inhibitor); dactolisib, PF-04691502, apitolisib, and omipalisib (PI3K inhibitors); and refametinib, binimetinib, trametinib, and pimasertib (MEK inhibitors). One phase I study of trametinib (MEK inhibitor) was terminated at the sponsor’s discretion. All other compounds tested in this study only reached the stage of phase I or II trials. 

### 2.3. Comparison of In Vitro Drug Testing and Clinical Trial Responses 

The response rate for each anticancer compound was calculated from our drug screen using DSS ≥ 5 as the cut-off point (Appendix A and Appendix A). We also collected the monotherapy objective response rate (ORR) from the clinical trials (Table 2). Based on the clinical trials, the ORR for cetuximab fell between 3.7% and 14.5% and the pooled ORR obtained using the inverse variance method for all eight clinical trials was 10.0% (confidence interval, CI 7.0–14.0; Figure 2A, Table 2). Cells cultured on top of (2D) or embedded (3D) in Myogel showed response rates closest to the clinical trials, at 17% (CI 4.4–37.8) and 25% (CI 5.0–49.5) across 12 HNSCC cell lines, respectively. Other testing conditions resulted in very high response rates to cetuximab: 67% (CI 40.0–93.3) of cells on plastic, 67% (CI 40.0–93.3) of cells on top of, and 75% (CI 50.2–99.5) of cells embedded in Matrigel. The predictive value of Myogel was similar to that for other EGFR inhibitors, such as afatinib and gefitinib (Figure 2B,C). Gefitinib showed only a weak response in four clinical trials with a pooled ORR of 6% (CI 3.0–9.0; Figure 2B). Cells cultured both on top of or embedded in Myogel showed a weak response to gefitinib at 8.3% (CI 7.3–24.0) across 12 HNSCC cell lines. Other testing conditions resulted in higher response rates: 16.7% (CI 4.4 to 37.8%) of cells on plastic, 41.7% (CI 13.8–69.6) of cells on top of, and 50% (CI 21.7–78.3) of cells embedded in Matrigel (Figure 2B). Based on three clinical trials of afatinib, the pooled ORR was 10% (CI 7.0–13.0; Figure 2C). Cells on top and embedded in Myogel had a low response rate of 17% to afatinib (CI 4.4–37.8) unlike other conditions, which all exhibited a high response rate of 75% (CI 50.2–99.5; Figure 2C).

Rapamycin analog mTOR inhibitors, such as temsirolimus which is currently in a phase II trial, showed a high response rate, but a relatively low effect (0 ≥ DSS < 10) in each of our in vitro testing conditions (Appendix A). Several DSS values were close to the cut-off point (DSS ≥ 5). The response rate for temsirolimus reached 75% (CI 50.2–99.5) on plastic and embedded in Matrigel, 67% (CI 40.0–93.3) on top of Matrigel, 83% (CI 62.2–104.4) on top of, and 50% (CI 21.7–78.3) embedded in Myogel (Figure 2D and Appendix A). A phase II clinical trial for temsirolimus resulted in promising tumor shrinkage in 39.4% of patients, but unfortunately lacked any objective response (Figure 2D and Table 2) [15]. Another phase II clinical trial reported only a 2.5% ORR (Figure 2D and Table 2). The pooled ORR for temsirolimus was 2% (CI 0–10.0). One window-of-opportunity trial for sirolimus (phase I and II trials) showed significant clinical responses with an ORR of 25% including one complete response (Table 2). The in vitro response rate for sirolimus reached 92% (CI 76.0–107.3) on plastic and on top of Matrigel, 67% (CI 40.0–93.3) embedded in Matrigel, 83% (CI 62.2–104.4) on top of, and 75% (CI 50.2–99.5) embedded in Myogel (Appendix A).

**Table 1 cancers-12-00092-t001:** Comparison of the culturing conditions for each anticancer compound. Adjusted *p*-values were collected, and the number and percentage of significant cases in the drug groups (EGFR, MEK, and mTOR/PI3K) were calculated. Significant differences appear in bold. (N/A, not applicable).

EGFR	Myogel 2D vs. Myogel 3D	Myogel 2D vs. control	Myogel 3D vs. control	Myogel 3D vs. Matrigel 2D	Myogel 3D vs. Matrigel 3D	Myogel 2D vs. Matrigel 3D	Myogel 2D vs. Matrigel 2D	Control vs. Matrigel 3D	Control vs. Matrigel 2D	Matrigel 2D vs. Matrigel 3D
Erbitux	1.000	0.118	0.081	0.081	**0.003**	**0.005**	0.118	1.000	1.000	1.000
Gefitinib	0.098	1.000	0.814	0.055	**0.012**	0.098	0.332	1.000	1.000	1.000
Erlotinib	1.000	1.000	1.000	0.454	0.142	0.142	0.454	1.000	1.000	1.000
Afatinib	1.000	0.118	0.118	**0.008**	**0.001**	**0.001**	**0.008**	1.000	1.000	1.000
Ganertinib	1.000	0.118	**0.019**	**0.002**	**0.003**	**0.024**	**0.019**	1.000	1.000	1.000
No. of sig. cases	0	0	1	2	4	3	2	0	0	0
%	**0.0**	**0.0**	**20.0**	**40.0**	**80.0**	**60.0**	**40.0**	**0.0**	**0.0**	**0.0**
**MEK**										
Pimasertib	1.000	1.000	1.000	**0.012**	**0.000**	**0.000**	**0.005**	**0.008**	0.282	1.000
Trametinib	1.000	0.707	1.000	**0.030**	**0.000**	**0.000**	**0.003**	**0.019**	0.707	1.000
Refametinib	1.000	1.000	1.000	**0.012**	**0.000**	**0.000**	**0.008**	**0.019**	0.707	1.000
Binimetinib	1.000	1.000	1.000	**0.004**	**0.000**	**0.000**	**0.004**	**0.005**	0.389	1.000
TAK-733	1.000	0.707	1.000	**0.012**	**0.000**	**0.000**	**0.003**	**0.019**	0.528	1.000
Selumetinib	1.000	1.000	0.707	**0.004**	**0.000**	**0.002**	**0.019**	0.169	0.814	1.000
No. of sig. cases	0	0	0	6	6	6	6	5	0	0
%	**0.0**	**0.0**	**0.0**	**100.0**	**100.0**	**100.0**	**100.0**	**83.3**	**0.0**	**0.0**
**mTOR/PI3**K										
Everolimus	1.000	1.000	1.000	0.142	**0.010**	**0.008**	0.118	0.612	1.000	1.000
Temsirolimus	N/A	N/A	N/A	N/A	N/A	N/A	N/A	N/A	N/A	N/A
Sirolimus	N/A	N/A	N/A	N/A	N/A	N/A	N/A	N/A	N/A	N/A
Ridaforolimus	N/A	N/A	N/A	N/A	N/A	N/A	N/A	N/A	N/A	N/A
Dactolisib	N/A	N/A	N/A	N/A	N/A	N/A	N/A	N/A	N/A	N/A
Apitolisib	1.000	0.707	1.000	0.528	0.067	**0.008**	0.098	1.000	1.000	1.000
Omipalisib	1.000	0.814	1.000	**0.201**	**0.004**	**0.000**	**0.012**	0.067	1.000	1.000
PF-04691502	1.000	1.000	1.000	0.332	**0.008**	**0.003**	0.169	0.332	1.000	1.000
No. of sig. cases	0	0	0	0	3	4	1	0	0	0
%	**0.0**	**0.0**	**0.0**	**0.0**	**37.5**	**50.0**	**12.5**	**0.0**	**0.0**	**0.0**

**Table 2 cancers-12-00092-t002:** Objective response rates in clinical trials of anticancer compounds used as monotherapies in head and neck cancer patients. Data collected from https://clinicaltrials.gov.

	Clinical Trial Number	Total Enrollment	Phase	Completion Year	Monotherapy Treated Patients	Responded Patients	Evaluation Criteria *	ORR% *	Notes
**Afatinib**	NCT01345682	483	3	2016	322	33	RECIST 1.1	10.2	
	NCT00514943	124	2	2013	62	5	RECIST 1.0	8.1	ORR is based on independent central review (ICR)
	NCT01415674	61	2	2006	41	3	RECIST1.1	7.3	Neoadjuvant treatment
**Gefitinib**	NCT00206219a [16]	486	3	2007	158	4	RECIST	2.7	Drug dose 250 mg/day
	NCT00206219b [16]				166	10	RECIST	7.6	Drug dose 500 mg/day
	NCT00015964 [17]	51	2	2005	47	5	N/A	10.6	
	NCT01185158 [18]	70	2	2004	70	1	RECIST	1.4	
	NCT00519077	44	2	2013	44	3	RECIST	6.81	
**Cetuximab**	NCT01040832	107	2	2012	53	3	RECIST 1.0	5.7	
	NCT00671437	42	2	2015	27	1	RECIST 1.0	3.7	ORR is based on CT scans
	NCT00661427a	61	2	2012	30	4	RECIST	13.3	Drug dose 500 mg/m^2^
	NCT00661427b				19	2	RECIST	10.5	Drug dose 750 mg/m^2^
	NCT00514943	124	2	2013	62	6	RECIST 1.0	9.7	ORR is based on independent central review (ICR)
	NCT01602315	27	2	2016	35	2	RECIST 1.1	5.7	
	NCT00939627	55	2	2014	22	1	RECIST 1.1	4.5	
	NCT01577173	122	2	2015	62	9	RECIST 1.1	14.5	
	NCT01696955	79	2	2017	38	3	RECIST 1.0	7.9	
**Temsirolimus**	NCT01172769 [15]	42	2	2012	33	0	RECIST	0	
	NCT01256385	86	2	2013	40	1	RECIST 1.0	2.5	
**Sirolimus**	NCT01195922 [19]	37	1 & 2	2015	16	4	RECIST 1.1	25.0	Neoadjuvant treatment

* Objective response rate (ORR) based on Response Evaluation Criteria in Solid Tumors (RECIST).

## 3. Discussion

Traditionally, new anticancer compounds are tested on cancer cell lines cultured on top of plastic wells, followed by testing using xenograft mouse models prior to clinical trials [1]. However, it is now clear that this strategy remains unsuccessful given the very low predictive value of the drugs’ efficacy in clinical trials [1]. We, thus, hypothesized that this failure is caused by missing the human physiologically relevant TME. Thus, we developed the first human-derived TME-mimicking matrix, “Myogel”, in which 66% of its protein content differs from the mouse tumor–derived Matrigel [2,3]. Thus far, we have demonstrated that a human tumor matrix induces the invasion of carcinoma cells more than mouse sarcoma–derived matrix “Matrigel” [2,3,4]. In this study, we compared traditional 2D plastic drug screening method with 2D and 3D Matrigel and Myogel. Combining data from monotherapy clinical trial results for HNSCC to our in vitro drug testing results, we demonstrated that Myogel improved the reliability of in vitro drug testing.

In solid tumors, cancer cells are influenced by TME containing hundreds of effectors, such as stromal and immune cells, their cytokines, growth factors, and structural matrix proteins [2]. Clearly, all of these factors interfere with the chemosensitivity of cancer against anticancer drugs [20]. Due to the unknown but presumably very large number of such proteins, it is quite difficult to precisely analyze the factors interfering in drug screening. Unfortunately, this also applies to our drug testing using Myogel representing the human TME.

Currently, cancer research is consistently moving towards 3D cell culture models for in vitro studies [20]. Thus, several matrices have been applied, such as Matrigel, collagen, and chemically defined synthetic hydrogels [21]. All of these matrices miss a broad spectrum of both human- and/or tumor-derived structural or signaling molecules. We, thus, hypothesized that a human tumor–derived matrix would provide a more appropriate TME for the cells and, therefore, a more in vivo–relevant drug response. 

Cell number measurement using imaging technique is not the optimal way to compare cell density in different matrices due to alternated cell morphology (spheroid in Matrigel and stellate-shaped in Myogel) and 3D growth (Appendix A). In this study, we used the CellTiter-Glo luminescent assay to determine the number of viable cells in culture plates. Even though some Matrigel wells showed low cell density under microscope, the luminescent assay readouts indicated similar or even larger number of viable cells compared to 2D and Myogel wells (Appendix A). This was because the cells on Matrigel are tightly packed together forming spheroid structures. We also did not find correlation between alternated cell growth in different matrices and the drug response.

The EGFR inhibitors were considered promising anticancer drugs for HNSCC. However, only modest response rates have been found [22]. Among all EGFR inhibitors, only cetuximab has been approved by the Food and Drug Administration (FDA) for HNSCC. Unfortunately, its impact has remained quite limited, and even responsive patients rapidly developed drug resistance [6,22]. Other EGFR inhibitors, such as gefitinib, have failed in phase III clinical trials despite their promising results in preclinical in vitro and in vivo animal studies [6,7,9,10,11]. Interestingly, in our setting, both plastic and Matrigel yielded higher response rates than the clinical outcomes of the EGFR inhibitors. This might explain why so many EGFR-targeted drugs successfully passed preclinical testing and subsequently failed during clinical trials [1]. Yet, EGFR inhibitors tested using Myogel wells exhibited similar response rates to the 16 monotherapy clinical trial results (with a total of 1258 patients).

In order to understand the mechanism behind the variable cell responses to EGFR and MEK inhibitors in different culturing conditions, we studied the protein expression of EGFR, ERK1/2, and pERK1/2 in five cell lines which revealed the most differential EGFR- and MEK-inhibitor effects between Myogel and the other testing conditions. After repeating the experiments three times, we could not see any clear difference in the protein levels of EGFR, ERK1/2, and pERK1/2 in these culture conditions. Therefore, there was no correlation between the protein levels and responses to EGFR and MEK inhibitors (Appendix A). Our in vitro results are in line with the in vivo results, which showed that EGFR protein expression and its gene copy number have failed as a predictive biomarker in numerous studies [12]. Additionally, previous in vitro study on HNSCC cell lines showed that EGFR amplification or overexpression was only weakly associated with EGFR inhibitor response [23]. While many studies report EGFR overexpression in HNSCC, supporting data are inconsistent and limited due to the large variation in antibodies, the lack of controls, and testing only at the RNA level [24]. One study revealed that EGFR protein overexpression more commonly occurs in established HNSCC cell lines (*n* = 14) than in clinical samples (*n* = 55) [25]. Clinical HNSCC samples (*n* = 55) did not overexpress EGFR at the protein level compared to healthy mucosa (*n* = 46) [25].

Several genomic alterations in HNSCC affect the PI3K/AKT/mTOR pathway activation [26], which plays an important role in cancer initiation and progression. mTOR inhibitors have shown promising anti-tumor activity in preclinical studies and early stage clinical trials in HNSCC [27]. Based on two phase II clinical trials, temsirolimus showed promising tumor shrinkage, but this was associated with no objective response [15]. Our in vitro results, relying on a DSS value of 5 as the cut-off point, did not predict patient outcome in clinical trials across all testing conditions. However, the majority of the tested cell lines yielded a low DSS value, close to the cut-off point of 5, which raises questions about the reliability of that score as a marker for an objective response. In one study, the authors only highlighted DSS values of less than 10 as non-responders [28]. If the cut-off point is increased to DSS > 10, the results more closely mirror patient responses. The selection of the most reliable response cut-off point is crucial and small changes in it could greatly induce the drug response rates, particularly when the DSS values are close to the cut-off point. Additionally, here we used only monotherapy clinical trials; those patients typically resistant to traditional treatment. This renders the comparison to the in vitro results relatively less than ideal. However, we excluded combination therapy trials, since separating the drug effect from other treatments (radiation or chemotherapy) would be impossible.

Another mTOR inhibitor, sirolimus, has thus far been studied in only one monotherapy HNSCC clinical trial among 16 patients. It showed an objective response rate of 25% and one complete patient response [19]. Although our in vitro study revealed a much higher response rate for sirolimus, further clinical trials are needed to interpret the in vitro results. 

Clearly, those drugs which target receptor activities, such as EGFR, are more greatly affected by the nature of the extracellular environment than those that target cytosolic enzymes, such as mTOR. This could explain Myogel’s ability to reveal the real response rate for EGFR antibodies better than for mTOR inhibitors. 

We predicted that a 3D culture would provide more reliable drug testing results than 2D monolayers. However, in contrast, 2D Myogel- and Matrigel-coated wells yielded rather similar results to 3D cultures for most of the drugs tested. Thus, our data suggest that a 2D-coated culture is suitable for drug testing purposes as long as the culture contains critical elements of the human TME. 

In conclusion, since the human tumor matrix improved the predictability of the in vitro anticancer drug testing of HNSCC cell lines, we argue that using it would reduce the number of false-positive preclinical results, the cost of drug development, and the unnecessary suffering of cancer patients.

## 4. Materials and Methods 

### 4.1. Cell Lines and Anticancer Compounds

We selected 12 of 45 HNSCC cell lines previously tested against 220 anticancer compounds on plastic (Appendix A) [23]. Each cell line was human papillomavirus (HPV)-negative and had wild-type KRAS. The cell lines were established at the Department of Otorhinolaryngology—Head and Neck Surgery, Turku University Hospital (Turku, Finland) [29]. Our selected cells included both primary and metastatic cell lines from different locations of the head and neck region. Cells were also selected based on their response to EGFR, MEK, and mTOR/PI3K inhibitors by taking both responsive and resistant cell lines. Additionally, we selected 19 effective or non-effective anticancer compounds, targeting the EGFR, PI3K-mTOR, and MAPK signaling pathways based on previous drug testing results (Appendix A) [23]. We cultured the cell lines in Dulbecco’s modified Eagle’s medium (DMEM)/F-12 (Gibco, 31330-038, Waltham, MA, USA) supplemented with 100-U/mL penicillin, 100-μg/mL streptomycin, 250-ng/mL fungizone, 50-μg/mL ascorbic acid, and 0.4-μg/mL hydrocortisone (all from Sigma Aldrich, St. Louis, MO, USA), and 10% heat-inactivated fetal bovine serum (FBS) (Gibco, 10270-106). All cell lines were mycoplasma-free, and tested using the PCR Mycoplasma Test Kit I/C (PromoKine, Heidelberg, Germany; cat no. PK-CA91-1048). Anticancer compound concentrations were selected based on a previous half maximal inhibitory concentration (IC50) evaluation [28].

### 4.2. 3D Matrices and Culturing Conditions

We designed different culturing conditions (cells on plastic, on coated wells, and embedded in matrix; Appendix A) for each clear-bottom 384-well plate (Corning® #3707, Corning, NY, USA). We used the commercial mouse Engelbreth-Holm-Swarm (EHS) sarcoma matrix, Matrigel (Corning; cat no. 354234), and the human leiomyoma matrix “Myogel” (invented by our group) [2,3]. The use of human leiomyoma tissue was approved by the Ethics Committee of Oulu University Hospital (statement number 35/2014). All prospective liquid handling was performed using an automated reagent dispenser (BioTEK, MultiFlo™ FX, Winooski, VT, USA).

Myogel and Matrigel were thawed overnight on ice (4 °C). We pre-chilled pipette tips and other equipment in a freezer (–20 °C) and 384-well plates on ice. Matrigel and Myogel were diluted with a cell culture media to 500 µg/mL. Matrices (5 µL) were added to 384-well plates using an electronic pipette (VIAFLO II, Integra, Zizers, Switzerland) and the plates were centrifuged for 2 min at 300 rpm at 4 °C. Plates were left overnight in a cell culture incubator. In the control plates, only the cell culture media was added. On the following day, the cells were counted using the Scepter™ 2.0 Cell Counter (Merck Millipore, Burlington, MA, USA) and suspended to 500 cells/well. The cell density was selected based on previous publication and optimized to avoid confluence on the last day of the experiment [23]. To the plastic and Myogel- and Matrigel-coated wells, cells were seeded using an automated reagent dispenser (BioTEK, MultiFlo™ FX, 20 µL, 25 000 cell/mL, 500 cell/well). Matrices for 3D cell embedding were diluted to a 500-µg/mL protein concentration. Type I rat collagen (500 µg/mL; Corning^®^, cat no. 354236) and 1N sodium hydroxide were mixed with Myogel (500 µg/mL) to ensure complete gelation. For embedding, cells were first mixed with the matrix (Myogel + Collagen I or Matrigel) and cell suspension was added to the wells using an electronic pipette (Integra, VIAFLO II; 10 μL, 50 000 cells/mL, 500 cell/well). After a 1-h incubation, a culture medium (10 μL per well) was added on top of the solidified matrix and the plates were returned to the incubator to sit overnight.

### 4.3. Drug Sensitivity and Resistance Testing

We performed drug sensitivity and resistance testing (DSRT) on HNSCC cell lines cultured in different matrices. Each drug was tested over a 10,000-fold concentration range. We normalized the drug effect against positive (BzCl) and negative dimethyl sulfoxide (DMSO) control wells to calculate the dose–response curves for each drug in each cell line and matrices separately. To quantitatively profile alternate drug effects, we calculated the drug-sensitivity score (DSS) designed by the High Throughput Biomedicine Unit (HTB) at the Institute for Molecular Medicine Finland (FIMM). The DSS score has been described in several studies [13,23,28], and appears to improve the identification of drug response vulnerabilities in cell lines and among patients by capturing several parameters in a single metric. These parameters include IC50, the slope of the concentration curve, and the minimum and maximum responses [13,28].

Our DSRT protocol was adapted from a platform for leukemia cells used previously [13]. DSRT custom plates were designed at HTB FIMM in Helsinki, Finland. We used five different 10-fold concentrations for each anticancer drug in the replicates (Appendix A). In total, 100-μM Benzethonium chloride (BzCl) served as the positive cell-killing control and 0.1% DMSO served as the negative control. The drug plates, containing all drugs except cetuximab (Erbitux, Merck KGaA, Darmstadt, Germany), were stored in pressurized inert nitrogen gas–filled storage pods (Roylan Developments Ltd., Surrey, UK). Cetuximab dilutions were added to the drug plates using a liquid handler (Echo® 525 acoustic dispenser, Labcyte, San Jose, CA, USA). We used five 10-fold cetuximab concentrations: 0.005, 0.05, 0.5, 5.0, and 49.5 µg/mL (from 0.03 to 340 nM). These concentrations were selected based on literature and after optimization at the HTB unit [30,31,32,33]. Drugs were diluted with a cell culture media, added by the MultiFlo™ FX automated reagent dispenser, and placed on a plate shaker for 60 min at 1000 rpm. Diluted drugs were added onto the cell culture plates (10 µL/well) using a pipetting robot (Biomek FX, Beckman Coulter, Brea, CA, USA). The final volume of each well was 35 µL.

### 4.4. Cell Viability Assay

After 3 days, the CellTiter-Glo^®^ (CTG) 2.0 Luminescent Cell Viability Assay (Promega, Madison, WI, USA) was used to determine the efficacy of the drugs. We first imaged the plates using IncuCyte Live-Cell Imaging System (Sartorius, Göttingen, Germany) to observe cell density (Appendix A). We then transferred the plates to room temperature for 15 min and dispensed CTG to the assay plates using the MultiFlo™ FX automated reagent dispenser (30 µL/well). The plates were shaken for 10 min at 450 rpm and then centrifuged for 5 min at 1000 rpm. Complete lysis of the cells was observed under a microscope. The CTG signal was detected using the PheraStar plate reader (BMG Labtech, Ortenberg, Germany).

### 4.5. Data Analysis

The signal measured in the drug-treated wells was normalized against positive and negative controls for each condition and the quality of each plate was measured using the Z’ factor (Appendix A) [34]. For each condition, dose–response curves were drawn based on a percent inhibition of viability and the drug concentration (Appendix A). The drug screening data analysis including calculation of the DSS [13] was performed using a pipeline developed at FIMM. The DSRT pipeline is available at https://breeze.fimm.fi/. The pipeline and scripts used for the drug sensitivity profiles were developed using the R programming language (Appendix A) [35]. 

We conducted the Friedman test using the SPSS software program (version 25, 2017, IBM Corporation, Armonk, NY, USA) on each drug comparing the five culturing conditions (control, Myogel 2D, Myogel 3D, Matrigel 2D, and Matrigel 3D) to detect the differences in drug efficacy. Pair-wise comparisons were performed (SPSS Statistics, version 25, 2017) applying the Bonferroni correction for multiple comparisons [14].

### 4.6. Clinical Trial Data Collection

Clinical trial data for 19 anticancer compounds in HNSCC patients were collected until 17 March 2018, from a website maintained by the National Library of Medicine (https://clinicaltrials.gov) and PubMed. We included only completed clinical trials for drugs used as monotherapies for which the results were reported. We collected information on the objective response rates (ORRs, measured according to the response Evaluation Criteria in Solid Tumors, RECIST), the number of patients who received the monotherapy, and the number of responding patients in one table.

### 4.7. Meta-Analysis of Clinical Data

We conducted a meta-analysis in order to graphically present the ORRs for the clinical trials collected, comparing them to the response rates obtained in this study under the five culturing conditions. In this analysis, we used RStudio, version 3.6.0. The R packages *meta*, *ggplot2*, and *ggthemes* were used for more specific analyses and for plotting. The ORRs and the confidence intervals were calculated using the *metaprop* function from the *meta* package, and the figures were obtained using the *ggplot* functionalities.

### 4.8. Immunoblot Analysis of EGFR, ERK1/2 and pERK1/2 Expressions in Growing Cells on Plastic, Matrigel and Myogel

To examine protein expression, 7.5 × 10^5^ cells were seeded on 25 cm^2^ flasks without coating or coated with 1-mg/ml Myogel and Martigel overnight at +37 °C. For the immunoblot analysis, we used five cell lines (UT-SCC-24A, UT-SCC-24B, UT-SCC-42A, UT-SCC-42B, UT-SCC-81), which revealed the differential EGFR- and MEK-inhibitor effects between Myogel and the other testing conditions (Appendix A). The cells were cultured for 48 h in a normal culture medium, after which they were washed three times with PBS and serum-free Opti-MEM (Gibco) was added. After 24 h, the cells were washed twice with PBS and lysed with an elution buffer (50 mM Tris HCl, pH 7.5, 10 mM CaCl2, 150 mM NaCl, 0.05% (*v*/*v*) Brij 35; Sigma Aldrich) with a complete protease inhibitor cocktail (Roche, Basel, Switzerland) and PhosSTOPTM phosphatase inhibitors (Roche). After removing the cell debris by centrifugation, the protein concentrations were measured using a DC Protein assay (Bio-Rad Hercules, CA, USA). Next, 30 µg of a soluble protein were separated under reducing conditions using a 12% sodium dodecyl sulfate polyacrylamide gel electrophoresis (SDS-PAGE) and transferred to an Immobilon P membrane (Millipore, Burlington, MA, USA). After blocking with the Odyssey^®^ Blocking Buffer (LI-COR Biosciences, Lincoln, NE, USA), the membrane was incubated with 1:1000 rabbit anti-EGFR (D38B1; Cell Signaling Technology, Danvers, MA, USA), 1:1000 rabbit anti-p44/42 MAPK (Erk1/2, 9102; Cell Signaling Technology), 1:2000 rabbit anti-Phospho-p44/42 MAPK (Erk1/2, Thr202/Tyr204, 9106; Cell Signaling Technology) or 1:2000 mouse anti-β-Actin (8226; Abcam, Cambridge, UK) antibodies overnight at +4 °C [36]. After washing three times for 5 min each with TBS-Tween^®^20 (0.05%), 1:10000 IRDye^®^ 680RD Goat anti-Rabbit IgG or IRDye^®^ 800CW Goat anti-Mouse IgG secondary antibody (Licor Biosciences) was added for 50 min at room temperature and the membrane was washed three times for 5 min each. The Odyssey scanner (LI-COR Biosciences) was used to image the membrane and we used the Fiji software [37] for quantification of the protein levels. The results were normalized to the corresponding β-Actin level (original blots shown in Appendix A).

## 5. Conclusions

Our findings taken together detected varying drug effects between the human tumor–derived matrix Myogel compared to current 2D and MSDM methods. We found that Myogel in in vitro drug screening replicates the best results from clinical trials. We suggest that using the human tumor matrix in preclinical studies will diminish the number of false-positive preclinical results, reduce the cost of drug development, and minimize the unnecessary suffering of cancer patients.

## Figures and Tables

**Figure 1 cancers-12-00092-f001:**
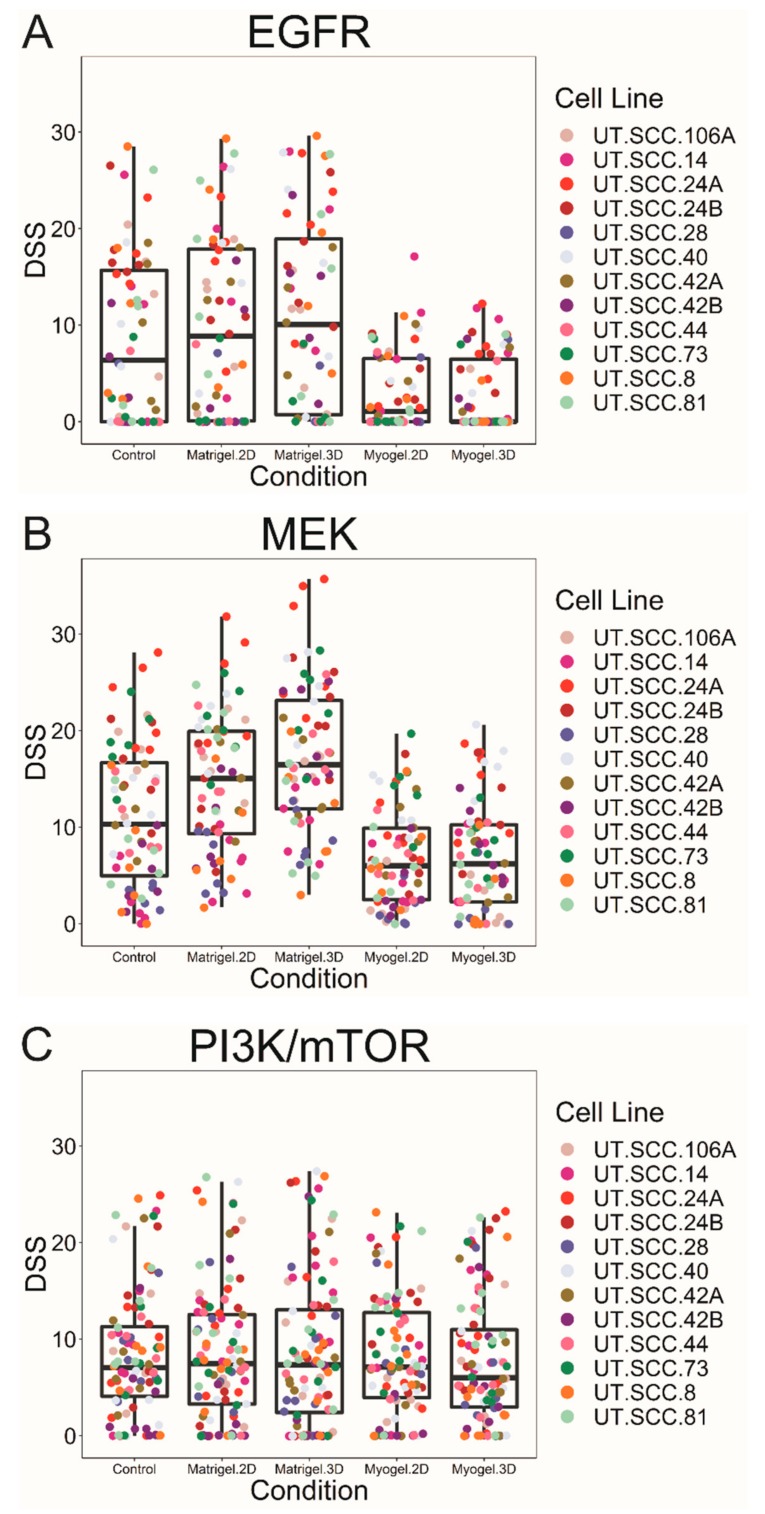
Box plots of the average drug-sensitivity score (DSS) values for each cell line clustered according to the drug group and the culturing condition. Cancer cells on top of and embedded in Myogel were less responsive to EGFR (**A**) and MEK inhibitors (**B**) compared to the cells cultured on two-dimensional (2D) plastic or Matrigel. However, for the PI3K and mTOR inhibitors (**C**), we observed a similar efficacy for the drugs under all culturing conditions.

**Figure 2 cancers-12-00092-f002:**
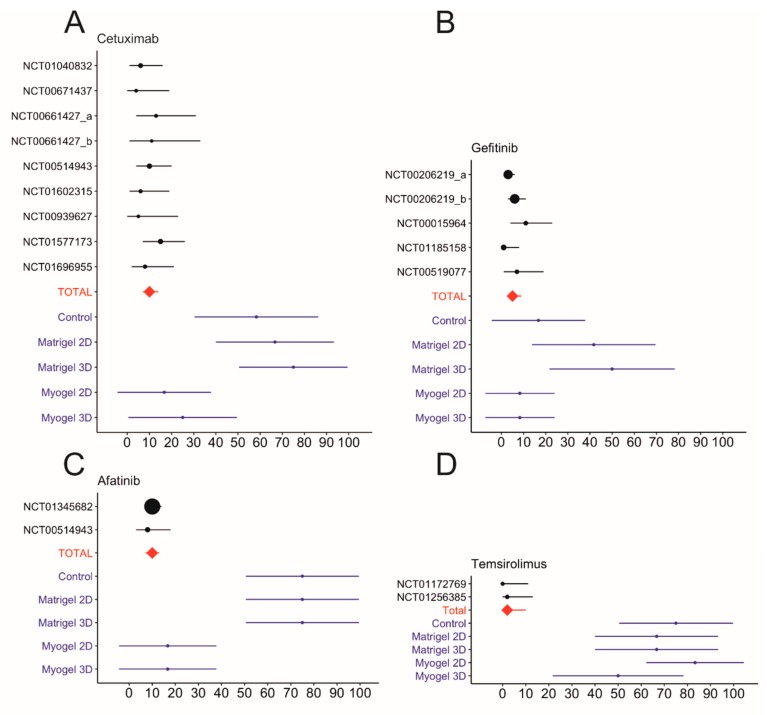
Meta-analysis of the objective response rates (ORRs) for cetuximab, gefitinib, afatinib, and temsirolimus monotherapy clinical trials and the response rates of in vitro drug testing. The pooled total average for the ORRs appears as a red diamond. The in vitro response rates were calculated using a drug-sensitivity score (DSS) value of ≥5 as the cut-off point (**A**–**D**). Data collected from https://clinicaltrials.gov. The different doses used in the same trial are marked as a and b.

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
