# Peer review of "Human Tumor–Derived Matrix Improves the Predictability of Head and Neck Cancer Drug Testing"

_cancers, 2019, doi:10.3390/cancers12010092_

Round 1

Reviewer 1 Report

Question (Q) 1a: The main concern is that it is difficult to understand what DSS stands for in reality because for example different cell lines have different plating efficiency and which probably also differ between different matrix.

Regarding different plating efficacy between matrices, we want to emphasize that plates were designed so that all matrices had its own negative and positive controls and drug responses were normalized using each controls.

Q 1b. It is well known that the density of cells has high effect on treatment response in vitro. Furthermore, the effect of drugs like Erbitux is inhibited or decreased proliferation and then 3 days is probably too short time to analyze the effect.

R 1b. Regarding cell seeding density, we seeded the same number of cells in each matrix, and the viability was measured for each cell line in all matrices. Cell density was selected according to other publications where adherent cancer cells were used and optimized to avoid confluence at the last day of the experiment (Lepikhova T et al. 2018). Cells had some alternations in viability in general (varied between cell line and matrix), but as pointed out before each matrix was normalized against its own negative and positive control.

Q 2a: Why is cells growing on Myogel more resistant for treatment?

Q 2b. Have cells a higher plating efficiency on Myogel?

R 2b: Regarding the plating efficiency, as we pointed above, the plates were designed so that each matrix has its own negative and positive control, and drug responses were normalized using each own control to eliminate effect of plating efficiency.

The author have not really answered my questions about plating efficiency. Their answer was that they used positive and negative controls is not the answer of my questions. Of course negative and positive controls have to be used. In Supplementary figure 6 the authors show 2 cell lines (UT-SCC-40 and UT-SCC-81) and the density of cells is much higher in images showing cells growing on Myogel 3D compare to cells growing on Matrigel 3D and platic. That is probably one explanation why cells growing on Myogel are less sensitive for cetuximab and other drugs compared to cell growing on Matrigel or plastic.

Next problem is when comparing cell lines with different density (and plating efficiency) using the method used in this manuscript.

Another question is the about the concentration of cetuximab (50 µg) used in Suppl. Figure 6. In the suppl. Table 3 the concentration for cetuximab was from 0.005 nM to 49,451 nM. What is the right concentration? 50µg is a very high concentration (332 nM) for in vitro studies. Most studies used concentrations between 15-80 nM. Why using such high concentration?

Q4: In this manuscript, the authors show results that 67-75% of the cell lines has a very high response rate to cetuximab. Other has published a high response rate to cetuximab of about 15 to 20% in cell panels of 25-30 cell lines. Explain why these cell lines under these condition are so more sensitive. Choice of cell lines, methods?

The authors answer for this question is not an explanation for the high response rate that not has been published before. Could the concentrations used have impact of the results or the method used?

Author Response

We thank the Reviewer for this valuable comments.

Reviewer 2 Report

I feel that authors satisfactory responded the reviewer's comments. Therefore, I recommend this paper for publication.

Author Response

We thank the Reviewer for her/his time and valuable positive feedback.

This manuscript is a resubmission of an earlier submission. The following is a list of the peer review reports and author responses from that submission.

Round 1

Reviewer 1 Report

The topic of the manuscript is of sufficient interest for the readers of the journal.

Originality

The topic is sufficient novel, important and interesting.

Structure

All in all, the article is clearly laid out, the structure is well organized. All key elements are present. The title exactly describes the content of the article.

Introduction

The introduction describes accurately, what the authors tried to achieve. The introduction is long enough, the topic is complex and the information given in the introduction summarizes the relevant clinical experience up to now rather well.

Results

The described results support the claims of the discussion part rather good. The section is quiet long but very interesting to read.

Imaging

The imaging material and graphs illustrate the required information from the text.

Discussion

The discussion sequence is good to read.

Author Response

We thank the Reviewer 1 for her/his time and valuable positive feedback.

Reviewer 2 Report

Cancers-584357
Title: Human tumor-derived matrix improves predictability of head and neck cancer drug testing
In this manuscript the authors have investigate the potential of using human tumor matrix (Myogel) in in vitro anti-cancer trials.
In conclusion, they proposed that cells grown on Myogel has the lowest expression of EGFR and yielded the highest similarity to response rate rates reported in EGFR inhibitors monotherapy clinical trials.
It is an interesting study, well designed but still with many weaknesses.
Comments and questions:
1. The main concern is that it is difficult to understand what DSS stands for in reality because for example different cell lines have different plating efficiency and which probably also differ between different matrix. It is well known that the density of cells has high effect on treatment response in vitro. Furthermore, the effect of drugs like Erbitux is inhibited or decreased proliferation and then 3 days is probably to short time to analyse the effect. Different drugs need different time points for drug response reading to get a reliable result.
2. Why is cells growing on Myogel more resistant for treatment? Have cells a higher plating efficiency on Myogel? Changes in EMT-profile?
3. Comparison between in vitro results and patient data is very difficult. In this manuscript, only patients that have got the treatment as monotherapy is included. The only groups with HNSCC that got for example Erbitux as monotherapy is patient that already has been treated with radiation or chemoradiation. So probably, these tumor cells has changed to a more resistant cell type compare to the tumor cells that has been used in vitro which has been established from untreated tumors. Therefore, the comparison done in this manuscript is far from perfect.
4. In this manuscript, the authors show results that 67-75% of the cell lines has a very high response rate to cetuximab. Other has published a high response rate to cetuximab of about 15 to 20% in cell panels of 25-30 cell lines. Explain why these cell lines under these condition are so more sensitive. Choice of cell lines, methods??
5. To confirm the multi drug screening results some experiments has to be performed where the authors can show the cell density, cell morphology, cell proliferation and cell death.
6. Figure 1 is very unclear. Try to show these results in more clear way.
7. Figure 2A. The quality is not good enough of the western blot. It´s looking better in the supplemental figure??
8. Figure 2B. Why are only 3 cell lines that not are used in the drug testing used for the EGFR western blot analyses? It has been published that the expression of EGFR can both decrease or increase when HNSCC cells are cultured in 3D were compared with 2D. Therefore, the EGFR expression of all cell lines need to be shown. How many experiments has been performed? The standard deviation has also to been shown.
9. It has been published that cetuximab high sensitive cell lines can have both low (similar or lower than normal oral keratinocytes) or high protein expression of EGFR. In the discussion, the authors explain the higher resistance to EGFR inhibitors with the decreased EGFR expression found in 3 cell lines cultured on Myogel. Furthermore, these 3 cell lines was not used in the multidrug screening. This has to be explained more.
10. More interesting when looking at the effect of EGFR inhibitors is activated EGFR (pEGFR). What happens with pEGFR when cells are cultured on Myogel?

Author Response

Question (Q) 1a: The main concern is that it is difficult to understand what DSS stands for in reality because for example different cell lines have different plating efficiency and which probably also differ between different matrix.

Response (R) 1a: We thank the Reviewer for this valuable comment.  The drug sensitivity score (DSS) has been designed in High Throughput Biomedicine Unit (HTB) at FIMM Helsinki, and used in several articles published in high quality journals e.g. Yadav B et al., Sci Rep 2014,  Pemovska T et al., Cancer Discov 2013, Lepikhova T et al., Mol Cancer Ther 2018. DSS score has been shown to improve the identification of cellular addictions and other vulnerabilities in individual cancer patients/cell lines over several parameters; IC50, slope of concentration curve and minimum and maximum responses (Yadav B et al., 2014). DSS value is almost equal to AUC (area under the curve) value which is commonly used in the drug discovery field. By automation, variation between wells has been reduced to minimum. DSS value makes comparison of cell lines, drugs and growing environment more robust compared to observation of IC50 values. As the method had been published and used in several articles, we added some more information about DSS score to the Method section and supplied the paragraph with references (Page 10, Line 325-344):

We performed drug sensitivity and resistance testing (DSRT) on HNSCC cell lines cultured in different matrices. Each drug was tested over a 10 000-fold concentration range. We normalized the drug effect against positive (BzCl) and negative (DMSO) control wells to calculate the dose–response curves for each drug in each cell line and matrices separately. To quantitatively profile alternate drug effects, we calculated the drug-sensitivity score (DSS) designed by the High Throughput Biomedicine Unit (HTB) at the Institute for Molecular Medicine Finland (FIMM). The DSS score has been described in several studies [13,23,28], and appears to improve the identification of drug response vulnerabilities in cell lines and among patients by capturing several parameters in a single metric. These parameters include IC50, the slope of the concentration curve, and the minimum and maximum responses [13,28].

Regarding different plating efficacy between matrices, we want to emphasize that plates were designed so that all matrices had its own negative and positive controls and drug responses were normalized using each controls. This is now described in the Data analysis section (Page 11, Line 362-364) as following: “The signal measured in the drug-treated wells was normalized against positive and negative controls for each condition and the quality of each plate was measured using the Z’ factor (Supplementary Table 4) [30].

Q 1b. It is well known that the density of cells has high effect on treatment response in vitro. Furthermore, the effect of drugs like Erbitux is inhibited or decreased proliferation and then 3 days is probably too short time to analyze the effect.

R 1b. Regarding cell seeding density, we seeded the same number of cells in each matrix, and the viability was measured for each cell line in all matrices.  Cell density was selected according to other publications where adherent cancer cells were used and optimized to avoid confluence at the last day of the experiment (Lepikhova T et al. 2018). Cells had some alternations in viability in general (varied between cell line and matrix), but as pointed out before each matrix was normalized against its own negative and positive control. We have now added the following text in Page 10, Line 312-313: “The cell density was selected based on previous publication and optimized to avoid confluence on the last day of the experiment [23].

Q 1c. Different drugs need different time points for drug response reading to get a reliable result.

R 1c. Regarding the duration of the experiment, we decided to use the same well documented standard procedure as applied previously in the high throughput unit, which allows drugs to affect 72 h (Yadav B et al., Sci Rep 2014, Pemovska T et al., Cancer Discov 2013, Lepikhova T et al., Mol Cancer Ther 2018). We fully agree with the Reviewer that 72 hours may not always be the most optimal for all of the drugs because of their different mechanism of action. However, as we tested mainly highly proliferative head and neck squamous cell carcinoma cell lines, the drugs´ effects on the cell proliferation was detectable using 72 h test period.

Q 2a: Why is cells growing on Myogel more resistant for treatment?

R 2: We thank the Reviewer for this important point and our main finding regarding the cells resistance to drugs in Myogel, which however, was very well in line with the in vivo clinical data of those drugs already used in the phase III trials. Unfortunately, we do not yet know the mechanism, and we have stated that fact in the manuscript (Page 8, Line 203-208) “In solid tumors, cancer cells are influenced by TME containing hundreds of effectors, such as stromal and immune cells, their cytokines, growth factors, and structural matrix proteins [2]. Clearly, all of these factors interfere with the chemosensitivity of cancer against anticancer drugs [20]. Due to the unknown but presumably very large number of such proteins, it is quite difficult to precisely analyze the factors interfering in drug screening. Unfortunately, this also applies to our drug testing using Myogel representing the human TME.

We have tried to understand the mechanism behind this resistance by conducting western blot for the EGFR, ERK1/2 and pERK1/2 but unfortunately the expression of these proteins were not correlated with the cells response to the anti-cancer drugs. This part is now added to the discussion Page 8, Line 226-236: “In order to understand the mechanism behind the variable cell responses to EGFR and MEK inhibitors in different culturing conditions, we studied the protein expression of EGFR, ERK1/2, and pERK1/2 in five cell lines which responded the most differentially to these inhibitors (Supplementary Figure 1). After repeating the experiments three times, we could not see any clear difference in the protein levels of EGFR, ERK1/2, and pERK1/2 in these culture conditions. Therefore, there was no correlation between the protein levels and responses to EGFR and MEK inhibitors (Supplementary Figure 5). Our in vitro results are in line with the previous in vivo results, which showed that EGFR protein expression and its gene copy number have failed as a predictive biomarker in numerous studies [12]. Additionally, previous in vitro study of used HNSCC cell lines showed that EGFR amplification or overexpression was only weakly associated with EGFR inhibitor response [23].

Q 2b. Have cells a higher plating efficiency on Myogel?

R 2b: Regarding the plating efficiency, as we pointed above, the plates were designed so that each matrix has its own negative and positive control, and drug responses were normalized using each own control to eliminate effect of plating efficiency.  

Q 2c: Changes in EMT-profile?

R2c: We agree with the Reviewer that Myogel might affect EMT-profile because its inducing invasion for some cell lines (Salo 2015, Salo 2018, Naakka 2019),  but this may not totally explain the reason behind drugs resistance on cells cultured on top of Myogel.

The effects of Myogel on cancer cells migration and invasion is stated in Page 2, Line 58-59: “Carcinoma cells show a markedly higher migration and invasion ability on Myogel compared to Matrigel [2-4]”

Q 3: Comparison between in vitro results and patient data is very difficult. In this manuscript, only patients that have got the treatment as monotherapy is included. The only groups with HNSCC that got for example Erbitux as monotherapy is patient that already has been treated with radiation or chemoradiation. So probably, these tumor cells has changed to a more resistant cell type compare to the tumor cells that has been used in vitro which has been established from untreated tumors. Therefore, the comparison done in this manuscript is far from perfect.

R 3: We thank the Reviewer for this important notice and we totally agree that this comparison is not ideal, but unfortunately, so far we know, this is the only applicable method to compare between in vitro drug testing and in vivo clinical trials. We agree that these patients might get this monotherapy as the last treatment option, but we could not use other trials where patients received a combination therapy, as it would have been impossible to separate drug´s effect from the other treatments (radiation/chemotherapy). We have now added this notice to the Discussion in Page 9, Line 257-261: “Additionally, here we used only monotherapy clinical trials; those patients typically resistant to traditional treatment. This renders the comparison to the in vitro results relatively less than ideal. However, we excluded combination therapy trials, since separating the drug effect from other treatments (radiation or chemotherapy) would be impossible.

Q4: In this manuscript, the authors show results that 67-75% of the cell lines has a very high response rate to cetuximab. Other has published a high response rate to cetuximab of about 15 to 20% in cell panels of 25-30 cell lines. Explain why these cell lines under these condition are so more sensitive. Choice of cell lines, methods?

R 4:  We selected all the cell lines in this study based on a previously screened large cell panel of head and neck squamous cell carcinoma cell lines (n=45, Lepikhova T et al. 2018). In Lepikhova screen, cell lines showed activity of EGFR inhibitors (afatinib, canertinib, erlotinib, and gefitinib) with median DSS > 10. In our study, we tried to select responsive and resistant cell lines to EGFR, MEK and mTOR/PI3K inhibitors. Unfortunately, cetuximab was not included in the previous drug panel by Lepikhova et al. (2018), and we had no previous data nor explanation concerning the response rate of these selected cell lines to cetuximab. We have now added the selection method into Materials and methods section (Page 10, Line 285-287): “Cells were also selected based on their response to EGFR, MEK and mTOR/PI3K inhibitors by taking both responsive and resistant cell lines.

Q 5: To confirm the multi drug screening results some experiments has to be performed where the authors can show the cell density, cell morphology, cell proliferation and cell death.

R 5: We have imaged the cell plates before the end-point viability assay using Incucyte automatic imaging system from Sartorius. However, cell density measurement by imaging is not an optimal way to compare cells growing in different matrices, because cells have alternated morphology and growing in different dimensions as shown and mentioned in the Supplementary Figure 3: “Cells were cultured in 2D plastic wells, on top or embedded in Matrigel or Myogel. Within Matrigel, UT-SCC-8 cells formed isolated round-shaped spheroids, whereas within Myogel cells were stellate-shaped.”

As suggested by the Reviewer, we now collected images of the two representative cell lines (UT-SCC-40 and UT-SCC81) showing cell density and morphology after 72 hours of the drugs treatment. New Figure is as a Supplement Figure 7. The following text is added to Page 11, Line 352-354 “We first imaged the plates using IncuCyte Live-Cell Imaging System (Sartorius, Göttingen, Germany) to observe cell density (Supplementary Figure 6).

For cell death analyses, our original plan was to use CellTox™ Green Cytotoxicity Assay to detect apoptosis. However, we tried this method for our 3D cultures, but failed due to high background signal from the matrices. It’s also known that apoptosis assay using CellTox™ Green is not optimal for highly proliferating cells, or cells which have leaky cell membrane.

Q 6: Figure 1 is very unclear. Try to show these results in more clear way.

R 6: We have shown Figure 1 data in different way in Supplementary figure 1, as a heat map.

Q 7: Figure 2A. The quality is not good enough of the western blot. It´s looking better in the Supplemental Figure ??

R 7: Because of the new western blot that we run for five more cell lines included in the drug testing, Figure 2A is now deleted, and the new western blot data is presented in new Supplementary Figure 5.

Q 8: Figure 2B. Why are only 3 cell lines that not are used in the drug testing used for the EGFR western blot analyses? It has been published that the expression of EGFR can both decrease or increase when HNSCC cells are cultured in 3D were compared with 2D. Therefore, the EGFR expression of all cell lines need to be shown. How many experiments has been performed? The standard deviation has also to been shown.

R8: Thank you for this valuable comment. We agree with the Reviewer that published EGFR expression data is not consistent and it has failed as predictive biomarker in numerous studies (as mentioned in Page 8 Line 233-234 “Our in vitro results are in line with the in vivo results, which showed that EGFR protein expression and its gene copy number have failed as a predictive biomarker in numerous studies [12].”

We agree that 3 cell lines not used for drug testing was insufficient and we performed more analysis using more cell lines (used in drug screen). We used those five cell lines (UT-SCC-24A, UT-SCC-24B, UT-SCC-42A, UT-SCC-42B, UT-SCC-81) which had the most variable responses to EGFR and MEK inhibitors on different matrices. We detected EGFR, ERK1/2, and pERK1/2 expression. Data is now collected in Supplementary Figure 5. Based on densitometry readings and normalized values, we could not find any correlation between drug response and EGFR, ERK1/2, and pERK1/2 expression. We added this section in discussion in Page 8, Line 226-236: “In order to understand the mechanism behind the variable cell responses to EGFR and MEK inhibitors in different culturing conditions, we studied the protein expression of EGFR, ERK1/2, and pERK1/2 in five cell lines which revealed the most differential EGFR- and MEK-inhibitor effects between Myogel and the other testing conditions. After repeating the experiments three times, we could not see any clear difference in the protein levels of EGFR, ERK1/2, and pERK1/2 in these culture conditions. Therefore, there was no correlation between the protein levels and responses to EGFR and MEK inhibitors (Supplementary Figure 5). Our in vitro results are in line with the in vivo results, which showed that EGFR protein expression and its gene copy number have failed as a predictive biomarker in numerous studies [12]. Additionally, previous in vitro study of used HNSCC cell lines showed that EGFR amplification or overexpression was only weakly associated with EGFR inhibitor response [23].” All technically successful western blots and densitometry readings including standard deviations are shown in the new Supplementary Figure 5. The results represent the average of one to three independent successful experiments.

Q 9: It has been published that cetuximab high sensitive cell lines can have both low (similar or lower than normal oral keratinocytes) or high protein expression of EGFR. In the discussion, the authors explain the higher resistance to EGFR inhibitors with the decreased EGFR expression found in 3 cell lines cultured on Myogel. Furthermore, these 3 cell lines was not used in the multidrug screening. This has to be explained more.

R9: We totally agree with the Reviewer that available studies showed that EGFR expression is neither predictive nor prognostic biomarker, and also the published in vitro data are not consistent. As suggested by the Reviewer we did a western blot for five more cell lines and unfortunately the new data did not support our previous claim so the text is now changed accordingly as clarified in the previous point (R8).

Q 10: More interesting when looking at the effect of EGFR inhibitors is activated EGFR (pEGFR). What happens with pEGFR when cells are cultured on Myogel?

R10: We tried to detect activated EGFR (pEGFR) several times using all five cell lines, but unfortunately we could not get any visible signal.

Reviewer 3 Report

In this study, authors examined the effect of anti-cancer drugs in head and neck squamous cell carcinoma (HNSCC) cell lines by using Myogel, compared to pastic and Matrigel. Authors found that cancer cells on Myogel responded less to EGFR and MEK inhibitors. Authors suggest that a human tumor matrix improves the predictability of in vitro anti-cancer drug testing compared to conventional methods, such as 2D and MSDM. 

I feel that this paper contains interesting findings.  Before acceptance, I have some comments as the following;

Authors should explain the detail of Myogel in "introduction". To use Myogel for anti-drug test, how do you always provide a same production lot? In Fig. 2, authors should check MEK and mTOR/PI3K.

Author Response

We thank Reviewer 3 for her/his time and valuable comments.

Q 1: Authors should explain the detail of Myogel in "introduction".

R 1: As requested by the Reviewer, we added more details about Myogel in Introduction section Page 2, Line 53-58: “Myogel provides a 3D culture environment for in vitro cancer studies, preserving the soluble factors present in the human TME by including, for example, soluble cytokines and growth factors [3]. Myogel proteome differs greatly from mouse sarcoma–derived Matrigel [3]. We previously demonstrated that 66% of the Myogel protein content differs from Matrigel [3]. Yet, these matrices share several proteins such as laminin, type IV collagen, heparan sulfate proteoglycans, nidogen, and epidermal growth factor [3].

Q 2: To use Myogel for anti-drug test, how do you always provide a same production lot?

R 2: As other natural animal derived matrices for example rat tail collagen and sarcoma–derived Matrigel, also Myogel has variation between lots. To minimize lot variation, we have pooled together several myomas from different patients. Different lots are numbered by unique code. Myogel lot used throughout in this study, Lot 270117, is a combination of 14 different myomas (total lot volume 280 ml).

Q 3: In Fig. 2, authors should check MEK and mTOR/PI3K

R 3: We thank the Reviewer for this comment. As the Reviewer suggested, we checked also MEK-ERK1/2 pathway. We revealed no clear difference in the ERK1/2 and pERK1/2 protein levels depending on the culturing conditions (Supplementary Figure 5). The new data is mentioned in the discussion Page 8, Line 226-236: “In order to understand the mechanism behind the variable cell responses to EGFR and MEK inhibitors in different culturing conditions, we studied the protein expression of EGFR, ERK1/2, and pERK1/2 in five cell lines which revealed the most differential EGFR- and MEK-inhibitor effects between Myogel and the other testing conditions. After repeating the experiments three times, we could not see any clear difference in the protein levels of EGFR, ERK1/2, and pERK1/2 in these culture conditions. Therefore, there was no correlation between the protein levels and responses to EGFR and MEK inhibitors (Supplementary Figure 5). Our in vitro results are in line with the in vivo results, which showed that EGFR protein expression and its gene copy number have failed as a predictive biomarker in numerous studies [12]. Additionally, previous in vitro study of used HNSCC cell lines showed that EGFR amplification or overexpression was only weakly associated with EGFR inhibitor response [23].

In case of mTOR/PI3K inhibitors, since cell lines showed similar response to mTOR/PI3K inhibitors regardless of the culturing condition (Figure 1 and Supplementary Figure 1), we decided not to investigate further the mTOR/PI3K expression using western blotting.